# Development of a conceptual framework for defining trial efficiency

**Charis Xuan Xie**[1]*, **Anna De Simoni**[1], **Sandra Eldridge**[1], **Hilary Pinnock**[2], **Clare Relton**[1]

**1** Wolfson Institute of Population Health, Queen Mary University of London, London, England, United Kingdom, **2** Usher Institute, Asthma UK Centre for Applied Research, The University of Edinburgh, Edinburgh, Scotland, United Kingdom

* charis.xie@qmul.ac.uk

## Abstract

### Background

Globally, there is a growing focus on efficient trials, yet numerous interpretations have emerged, suggesting a significant heterogeneity in understanding "efficiency" within the trial context. Therefore in this study, we aimed to dissect the multifaceted nature of trial efficiency by establishing a comprehensive conceptual framework for its definition.

### Objectives

To collate diverse perspectives regarding trial efficiency and to achieve consensus on a conceptual framework for defining trial efficiency.

### Methods

From July 2022 to July 2023, we undertook a literature review to identify various terms that have been used to define trial efficiency. We then conducted a modified e-Delphi study, comprising an exploratory open round and a subsequent scoring round to refine and validate the identified items. We recruited a wide range of experts in the global trial community including trialists, funders, sponsors, journal editors and members of the public. Consensus was defined as items rated "without disagreement", measured by the inter-percentile range adjusted for symmetry through the UCLA/RAND approach.

### Results

Seventy-eight studies were identified from a literature review, from which we extracted nine terms related to trial efficiency. We then used review findings as exemplars in the Delphi open round. Forty-nine international experts were recruited to the e-Delphi panel. Open round responses resulted in the refinement of the initial nine terms, which were consequently included in the scoring round. We obtained consensus on all nine items: 1) four constructs that collectively define trial efficiency containing scientific efficiency, operational efficiency, statistical efficiency and economic efficiency; and 2) five essential building blocks for efficient trial comprising trial design, trial process, infrastructure, superstructure, and stakeholders.

**Data Availability Statement:** All relevant data are within the manuscript and its supporting information files.

**Funding:** CX is funded by the Wellcome Trust (224863/Z/21/Z). URL: https://wellcome.org/. For

**Competing interests:** The authors have declared that no competing interests exist.

## Conclusions

This is the first attempt to dissect the concept of trial efficiency into theoretical constructs. Having an agreed definition will allow better trial implementation and facilitate effective communication and decision-making across stakeholders. We also identified essential building blocks that are the cornerstones of an efficient trial. In this pursuit of understanding, we are not only unravelling the complexities of trial efficiency but also laying the groundwork for evaluating the efficiency of an individual trial or a trial system in the future.

## Introduction

Worldwide, trial efficiency is a longstanding priority for the pharmaceutical industry [1], academia and funding bodies [2,3]. In 2004 in the US, the Clinical Trials Working Group of the National Cancer Advisory Board set the goal of improving operational efficiency to facilitate timely and cost-effective trial execution [4]. In the UK, the National Institute for Health and Care Research offers additional funding to support clinical trial units to advance the design and execution of efficient, innovative research, aiming to provide robust evidence to inform clinical practice and policy [5]. A recent article in The Lancet Global Health examined the challenges faced by current clinical trial research in low- and middle-income countries, and argued that efficient trials are needed to address research questions related to the increasing burden of non-communicable diseases in a timely and affordable way [6].

Currently, the concept of efficiency in healthcare trials has been used to refer to accelerated ethical approval [6], addressing multiple complex questions in a single trial [7] and with a minimised sample size [6], trials conducted with shorter duration [7,8], lower costs [9], and reduced resource requirements [10]. In addition, existing literature has discussed trial efficiency in terms of operational efficiency [11–13], scientific efficiency [11], statistical efficiency [13,14], and economic efficiency [15]. There is significant heterogeneity as to what is meant by efficiency in the context of trials, which may hinder effective communication and decision-making between stakeholders, and compromise the comparability of studies. Therefore, in this study we aimed to develop a conceptual framework for defining trial efficiency and to achieve expert consensus on the framework constructs.

## Method

### Study design

We undertook a literature review to identify items that define and comprise trial efficiency. We then conducted an e-Delphi study to refine and validate those items and to achieve consensus on the constructs and the building blocks of trial efficiency. The ethics approval was obtained from Queen Mary University of London research ethics committee (QMERC22.316). This study follows the Guidance on Conducting and Reporting Delphi Studies (CREDES) [16].

### Literature review for generating items

Our goal in the literature review was to collate existing discussions on efficiency in the context of trials, including definitions and attributes described as constituting an efficient trial. As discussions specifically focused on this subject are scarce, we included a broad range of study

types, such as full trial papers or protocols, editorials, and opinion pieces that discussed trial efficiency. We considered all types of human trials evaluating medical, surgical, or behavioural interventions, including efficacy trials, effectiveness trials, and implementation trials. The search was limited to English-language articles, and there was no restriction on publication dates. To carry out the review, we searched MEDLINE (via Ovid) database, for terms such as 'trial' and 'efficien*' in article titles and keywords. As 'efficiency' is a common word in literature, we searched for these two keywords only within article titles (rather than within the abstracts) ensuring the results' relevance to the discussion of trial efficiency. The detailed inclusion and exclusion criteria are listed in S1 Table.

### e-Delphi

**Panel selection and recruitment.** The aim was to recruit a diverse panel of experts from the trial community, encompassing a range of roles and perspectives. This included international researchers identified through the literature review, colleagues who are part of professional trial networks such as UK trial managers' network, representatives from funding bodies, journal editors, and members of the public who have been involved in trials. Purposive sampling and snowball sampling methods were then used to identify additional participants. We approached those participants with known contact details by individual emails generated through Clinvivo [17], while for colleagues within professional networks, where we didn't have individual contact details, we sent a generic recruitment email to the network's mailing list. Recruitment began in November 2022 and continued until March 2023. Written informed consent was obtained online through the Clinvivo Delphi system.

**Data collection.** We opted for two rounds of data collection because consensus was achieved by the end of the second round. These rounds were preceded by a pilot round to test the feasibility of the open round.

*Pilot test.* We pilot tested the feasibility of the open round questionnaire amongst colleagues with diverse experience in trial design and conduct at the Pragmatic Clinical Trial Unit of Queen Mary University of London. This provided valuable feedback on the clarity of the questions, the appropriateness of the response options, and the overall structure of the questionnaire. Based on the feedback received during the pilot testing, we made revisions and refinements to the questionnaire to enhance its usability.

*Open round.* In the open round, we invited panellists to share their thoughts on 1) their understanding of trial efficiency and 2) the most efficient or inefficient aspects they have encountered in the trials they have conducted or in which they have participated. These questions were designed as free-text to encourage detailed, narrative responses. To gain insights into the participants' backgrounds, we collected information on countries of residence, and roles within the trials (see S1 File for the questionnaire). This open round allowed us to gather diverse viewpoints and experiences related to trial efficiency which contributed to the development of a comprehensive set of items for ranking in the subsequent round. The data collection for this round took place over four weeks, with reminder emails sent to participants after the second and third weeks.

*Scoring round.* Panel members from open round were emailed a link to the second questionnaire. They were asked to rate the importance of the proposed items on a scale of 1 to 9 (1: not at all important to 9: critically important). At the end of each question, there was a free text space for any comments they wished to share. The scoring round data collection spanned four weeks with weekly reminders to participants.

**Data analysis and consensus.** Descriptive statistics were used to analyse quantitative demographics and thematic analysis was used to summarise free text responses from both

Delphi rounds. To assess disagreement and appropriateness, we used the Research ANd Development (RAND)/ University of California Los Angeles (UCLA) appropriateness method [18]. It involves calculating the median score, the inter-percentile range (IPR) (30th and 70th), and the inter-percentile range adjusted for symmetry (IPRAS), for each item being rated. Consensus was defined as items rated "without disagreement", measured by the IPRAS.

**Patient and public involvement.** In this study, members of the public (n = 4) (including two who had participated in trials) were invited to share their thoughts, participate in the ranking process, provided with the outcomes of each round upon completion. They were considered experts due to their lived experience and offered £30 voucher as a compensation for their time.

## Results

### Delphi participants

Out of 106 international experts approached, and 4 e-mails sent to network mailing lists, forty-nine participants responded to the open round (United Kingdom (n = 37), United States (n = 7), Canada (n = 2), Australia (n = 1), Ireland (n = 1), and Kenya (n = 1)). The panel included a diversity of roles including statisticians (n = 17), trial managers (n = 12), principal investigators (n = 7), funders (n = 4), journal editors (n = 3), member of the public (n = 4), data managers (n = 3), site staff (n = 2), sponsors (n = 2), researchers (n = 2), monitors (n = 2), ethicist (n = 1), clinician (n = 1), CTU manager (n = 1), trial support officer (n = 1), and trial methodologist (n = 1). Many participants had more than one role. See Fig 1.

### Literature review

We included a total of 78 studies for data analysis (see S1 Fig), including 6 (8%) reviews, 15 (19%) perspectives or commentaries, 1(1%) interview, 2 (3%) case studies, 2 (3%) surveys and 3 (4%) randomised trials, and 49 (63%) methodologies describing new trial designs. Only 8 (10%) studies had explicitly defined or explained what 'efficiency' meant in the context of their trials (see S2 Table for details). We categorised discussions of efficiency from the literature into

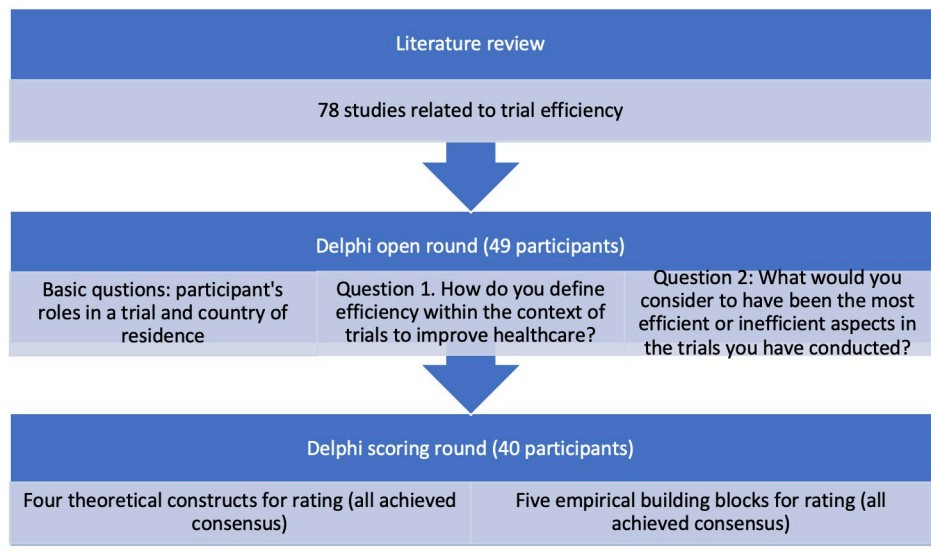

**Fig 1. Delphi flowchart.**

**Table 1. Key themes synthesised from literature review.**

| How efficiency had been discussed | Examples and references |
|---|---|
| Scientific efficiency | Scientific efficiency refers to the methodological rigour of the trial design. That is, a design that uses fewer resources and less infrastructure to maximise the outputs [11], addresses the right research questions, considers the implications of the design decision, and is relevant to the stakeholders [19,20]. |
| Operational efficiency | Operational efficiency covers full trial processes, from concept development to protocol activation, from enrolment to closure stage [11]. Wu and colleagues [20] assessed operational efficiency in patient recruitment and trial duration, Hess and colleagues [21] increased operational efficiency through objective site selection and reduced site coordinator workload. In addition, the National Cancer Institute established the Operational Efficiency Working Group to identify barriers associated with trial operations, aiming to reduce trial activation time and timely complete the activated studies [76]. |
| Statistical efficiency | Statistical efficiency measures the choices of estimators [24], experimental designs and hypothesis testing procedures [22], type I error, the power, and the sample size [23], the use of endpoint events include the selection of an appropriately weighted test statistic [14]. |
| Economic efficiency | Economic efficiency concerns the expenditure of research resources [15] and the cost for completing the trial [25]. |
| Trial designs | Including adaptive designs [23,26–33], master protocol trial designs [34] such as basket trials [35,36] and platform trials [37,38], sequential trial designs [7,39], clusters designs [40–42], factorial trials [43,44] and registry-based trials [45] |
| Trial conduct | 1) patient identification and recruitment [20,46–53], for example, the automated eligibility screening tool increased the efficiency of patient accrual. |
| | 2) data analysis [54–57], for example, "an alternative analytical approach that can enhance the signal-to-noise ratio would open the path for more efficient and rigorous clinical trials of Parkinson's Disease therapies". |
| | 3) selection of endpoints or outcome measures [58–61], for example, the use of ordinal outcomes and composing outcomes within a patient could improve trial efficiency. |
| | 4) data collection and management [21,62], for example, collecting and processing routine health data from the existing registry would facilitate efficient trial conduct. |
| | 5) site selection and management [21,63–65], such as reduced site workload and improved site operation contributed to trial efficiency. The central argument in this group was to improve trial efficiency by enhancing its operational efficiency [11,66]. |
| Other aspects | 1) using information technologies and mobile apps [53,67–70] |
| | 2) involving the public and stakeholders [20,71] |
| | 3) efficient trial reviews and regulatory approvals [28,66,72–74] |

nine key items: 1)scientific efficiency [11,19,20], 2)operational efficiency [11,20,21], 3)statistical efficiency [14,22–24] and 4)economic efficiency [15,25], 5)efficiency in trial designs [7,8,23,26–45], 6)trial conduct [11,20,21,46–66], and other aspects such as 7)improving efficiency using information technologies and mobile apps [53,67–70]; 8)involving the public and stakeholders [20,71]; and 9)efficient trial reviews and regulatory approvals [28,66,72–74]. (see Table 1 for details). These results were included as exemplars in the Delphi open round questionnaire. The detailed description of the literature review has previously been made available [75] to ensure full transparency and to facilitate open scholarly dialogue.

## Open round

When asked to define trial efficiency, some participants referred to definitions from the literature review, while other cited similar definitions tailored to their trial context. When asked about the most efficient/inefficient facets of trial efficiency, the responses resonated closely

with the findings from our literature review (Fig 2). Specifically, trial design emerged as the facet most frequently cited as enhancing efficiency, whereas data collection was often highlighted as the element that most impeded efficiency.

By incorporating findings from this round, we further refined the nine items identified from the literature review and divided them into two groups: 1) theoretical and abstract constructs: scientific efficiency, operational efficiency, statistical efficiency, and economic efficiency; 2) empirical and fundamental building blocks: trial design (including endpoints selection, statistical analysis plan, protocol development, etc.), trial process (including recruitment and retention, data collection and analysis, trial administration, etc.), superstructure (including regulatory approvals, funding application etc.), infrastructure (including financial and physical resources such as cost, information technologies, routine healthcare data, etc.), and stakeholders. This resulted in a total of nine items for rating in the scoring round (see Table 2).

## Scoring round and consensus

Forty participants responded (82%) to the scoring round and there was no disagreement on any items (Table 2). We also conducted sub-analyses by five role groups: (1) funders and sponsors (n = 6); (2) statisticians (n = 13); (3) trial managers (n = 10); (4) principal investigators (n = 6); and (5) PPIs (n = 3). Group membership was not mutually exclusive. Stratified results showed widespread agreement that the items were appropriate, with the exception of one of the building blocks–superstructure. The funders and sponsors group disagreed this item was appropriate (S3 Table). As a result, no new items were added but we slightly modified the explanation of each proposed item, in line with free-text comments made by the participants.

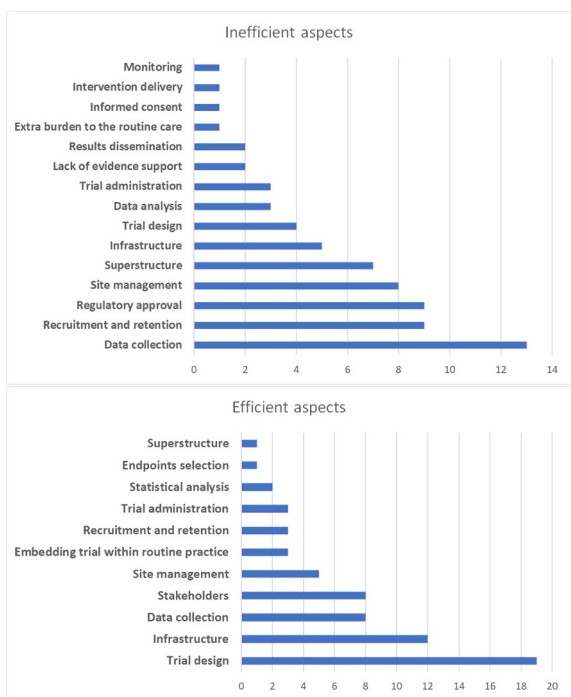

**Fig 2. The efficient and inefficient aspects discussed in the open round.** The x-axis represents the frequency of responses.

**Table 2. Scoring round items and results: Appropriateness, disagreement, median item ratings, interpercentile range, and intercentile range adjusted for asymmetry.**

| Item | Disagreement | Median | P30 | P70 | IPR | IPRAS |
|---|---|---|---|---|---|---|
| 1.1 Scientific efficiency: methodological rigour of the trial design | No | 9 | 8 | 9 | 1 | 7.6 |
| 1.2 Operational efficiency: optimal management, organization, and execution of trial processes and procedures | No | 9 | 8 | 9 | 1 | 7.6 |
| 1.3 Statistical efficiency: a measure of quality of an estimator, of an experimental design, or of a hypothesis testing procedure | No | 8 | 7.5 | 9 | 1.5 | 7.225 |
| 1.4 Economic efficiency: optimal use of resources in the design, implementation, and analysis of clinical trials | No | 8 | 7 | 8.5 | 1.5 | 6.475 |
| 2.1 Trial design: planning and organisation of a trial | No | 9 | 9 | 9 | 0 | 8.35 |
| 2.2 Trial process: trial set up, conduct and closeout | No | 9 | 8 | 9 | 1 | 7.6 |
| 2.3 Stakeholders: individuals or groups who have an interest or concern in the design, execution, and outcomes of a trial | No | 8 | 7 | 9 | 2 | 6.85 |
| 2.4 Infrastructure: underlying framework, systems, and resources required to design, implement, manage, and analyse a trial | No | 8 | 8 | 9 | 1 | 7.6 |
| 2.5 Superstructure: overarching structure of a trial | No | 8 | 7 | 8 | 1 | 6.1 |

$P_{30}$: inter-percentile range $30^{th}$.

$P_{70}$: inter-percentile range $70^{th}$.

IPR: inter-percentile range.

IPRAS: inter-percentile range adjusted for symmetry.

### Theoretical constructs of trial efficiency: Revised definitions incorporating Delphi comments

**Scientific efficiency.** Some participants were confused by the provided definition (Box 1. quote 1); while some suggested expanding the definition with the inclusion of feasibility and implementation (Box 1. quotes 2–3). As such, we refined the definition as the balance of methodological rigour, relevance of the research question, and feasibility of trial design. It prioritises effective use of resources, including data, to minimise research waste, considers the alignment of design and statistical strategies, and underscores the importance of the study's practical impact on stakeholders and delivering value to end-users.

**Operational efficiency.** Some comments suggested the definition should be expanded to consider operation feasibility, bureaucracy, and ongoing evaluation (Box 1. quotes 4–6). Therefore, we modified operational efficiency as the optimal management, organisation, execution, and continuous evaluation of trial processes and procedures. It emphasises operational feasibility (such as ensuring there are enough workforce, managing delays, and working effectively with third-party providers), reducing unnecessary bureaucracy and duplication, and continuously assessing the trial for potential improvements.

**Statistical efficiency.** The initial definition (Table 1) was expanded based on the participants' comments (Box1. quotes 7–8), as the application of design and analytical methods that result in more accurate estimates of treatment effects or other parameters of interest. This includes considerations of minimising the amount of data to be collected, accounting for missing data, and managing sources of bias or confounding; its focus is specifically on maximising the accuracy and reliability of results given the data collected.

**Economic efficiency.** We increased the clarity of the initial definition according to scoring round feedback (Box 1.quotes 9–10): the optimal use of resources in the trial design, implementation and analysis, to ensure immediate and long-term cost-effectiveness of the trial. This focus on value ensures that resources are utilised to their fullest extent without compromising the quality of the research. It emphasises on the cost-effectiveness of conducting the trial.

**Box 1. Scoring round exemplar free-text comments related to the construct definitions**

Scientific efficiency

*Quote 1*: *"Not sure rigour equates to efficiency"* (Participant n. 17, principal trial investigator)

*Quote 2*: *"Feasibility of trial design needs to be included here. You could have the perfect trial design but no participants or high withdrawals and lack of site engagement."* (Participant n.2, trial manager)

*Quote 3*: *"This may also need to include how important the findings will be to service users and the public and whether there are ways they are expected to be implemented in practice."* (Participant n.28, trial support officer)

Operational efficiency

*Quote 4*: *"I'd make particular focus on the bureaucracy - endless paperwork."* (Participant n.3, funder)

*Quote 5*: *"Feasibility of operational efficiency. You may have participants and engaged sites but you need operational feasibility to align."* (Participant n.2, trial manager)

*Quote 6*: *"Would like to see reference to the ongoing assessment of a trial in the descriptor."* (Participant n.39, trial manager)

Statistical efficiency

*Quote 7*: *"and accounting for missing data, and sources of bias or confounding"* (Participant n.19, principal trial investigator)

*Quote 8*: *"Also needs to encompass other aspects of analysis, e.g., health economics."* (Participant n.14, statistician)

Economic efficiency

*Quote 9*: *"Allowing for the concept of data sharing beyond the life of the study"* (Participant n.37, sponsor)

*Quote 10*: *"Need to be clear that this is (I presume) related to the costs of delivering the trial and not the cost of the intervention (i.e. health economic analysis)."* (Participant n.26, statistician)

## Essential building blocks comprising an efficient trial

Overall, there was a strong consensus on the building blocks; the free-text comments did not suggest significant alterations, but recommended adding some details within each building block. Trial design concerns the planning and organisation of a trial, which may include the trial methodologies, research questions, sample size, interventions, control group, endpoints and outcomes; document development such as funding application; as well as planning feasibility and pilot studies. The trial process involves the setup, execution, and closeout phases of a trial (see S2 Fig for details). Stakeholders are the critical human factor, they are individuals or groups with an interest or concern in the design, execution, and outcomes of a trial. They could be trial participants (e.g. patients, practitioners, health system leaders, public health organisations, etc.), trialists (e.g. investigators, researchers, trial managers, statisticians, etc), funders, sponsors, trial sites and their staff, regulatory authorities, healthcare and clinical practitioners, the scientific community (researchers, academics, and clinicians interested in the trial's outcomes and its implications for future research) and the general public (the broader population who may ultimately benefit from the knowledge generated by the clinical trial). Infrastructure is the underlying framework, systems, and resources required to design, implement, manage, and analyse a trial, such as resources (human, financial, physical), information systems and technologies, and healthcare data. Superstructure serves as the overarching structure of a trial, including laws, policy, and governance.

With these, we developed a Trial Efficiency Pentagon (Fig 3) to place the five building blocks and to illustrate the multiple connections among them - improvements in one block may potentially lead to trade-offs in one or more other blocks.

## The final conceptual framework for defining trial efficiency

Fig 4 represents the finalised framework. The term trial efficiency is complex and multifaceted, encompassing four conceptual constructs with five essential building blocks.

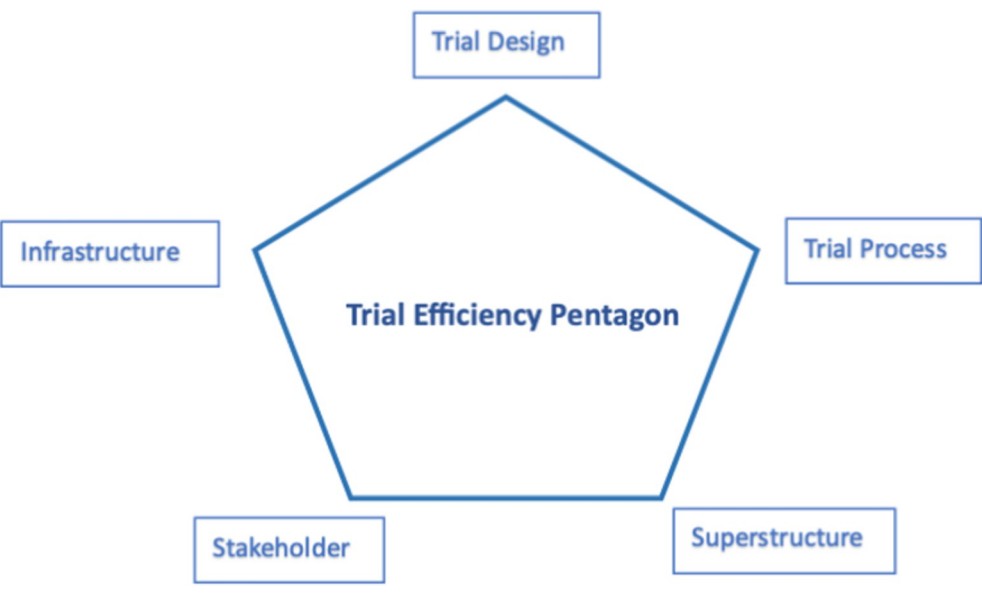

**Fig 3. Trial efficiency pentagon.**

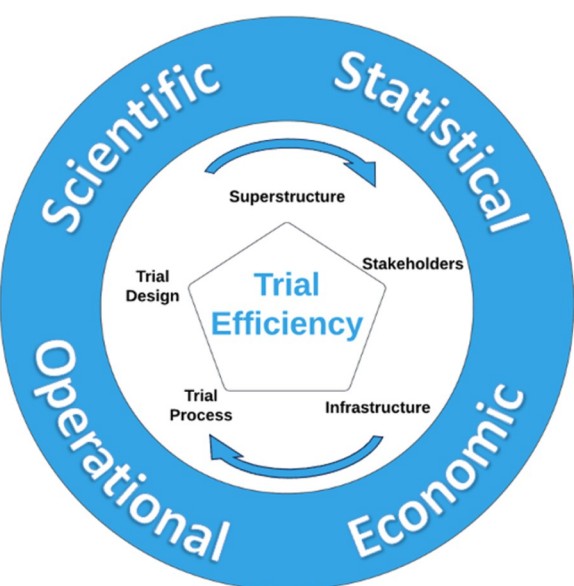

**Fig 4. The conceptual framework of trial efficiency.** The outer blue circle outlines theoretical constructs of trial efficiency: Scientific Efficiency, Statistical Efficiency, Operational Efficiency and Economic Efficiency. At its core, the inner pentagon outlines the empirical building blocks: Superstructure, Stakeholders, Infrastructure, Trial Process, and Trial Design. The cyclical arrows indicate the necessity for a balanced consideration of each building block within each construct to optimise trial efficiency.

## Discussion

### Main findings

Consensus was achieved on the four constructs that together define trial efficiency: scientific efficiency, operational efficiency, statistical efficiency and economic efficiency; and the five essential building blocks for considering an efficient trial: trial design, trial process, infrastructure, superstructure, and stakeholder.

### The conceptual constructs, empirical building blocks, and interrelationships

Overall there was no disagreement over the constructs that conceptually define trial efficiency. However, some concerns were raised regarding potential overlaps, between scientific efficiency and statistical efficiency, and between operational efficiency and economic efficiency (S4 Table). These four constructs share some common elements. However, they are conceptually distinct and each construct brings unique aspects to the concept of trial efficiency. Scientific efficiency, for instance, focuses primarily on the methodological rigour [77] and feasibility of trial design, while statistical efficiency is concerned with achieving the most accurate results possible with the smallest amount of data collected [78]. The overlap lies in the fact that both aim to optimize the quality and validity of the trial's findings, yet their distinct focus underlines their separate roles within the overarching construct of trial efficiency. Similarly, while operational and economic efficiency both aim to make the best use of resources [11], they do so in different ways and in different contexts. Operational efficiency is about the effective management and organization of trial processes and procedures [11,13], while economic efficiency involves optimizing resource use in relation to the cost of delivering the trial. By maintaining these conceptually distinct constructs, we were able to capture the broad spectrum of abstract

factors that define trial efficiency, thus offering a nuanced theoretical framework for its comprehension.

The proposed building blocks create a foundation for the formulation of an efficient trial. In the Delphi scoring round, there was strong consensus regarding the significance of these building blocks, with an average median score of 8.4 on a 1–9 scale. However, some participants perceived hierarchy among the building blocks, suggesting that some (e.g., trial design and process) hold more importance than others. This was reflected in the literature review and responses in the Delphi open round, where certain building blocks - such as trial design - were more frequently discussed as critical determinants of trial efficiency. Despite these observations, we propose that all five building blocks have equal importance and they mutually contribute to the overall efficiency of the trial. These foundational elements are also interconnected, for instance, even the most rigorous and feasible trial design is contingent upon the availability of suitable infrastructure support and requires inputs from stakeholders. Therefore, we advocate for a balanced view where no single building block takes precedence in the trial efficiency pentagon.

There is a layered connection between the constructs and the building blocks: the constructs were conceptualised to provide a broad, overarching view of efficiency within healthcare trials. In contrast, the building blocks were identified as the essential, practical components that operationalise efficiency in real-world settings. In addition to this relationship, we suggest that for a comprehensive understanding, each efficiency construct takes into account all five building blocks. For instance, while it may seem apparent that scientific efficiency is closely linked with trial design, focusing on how the study is conceptualised to ensure methodological soundness; it also intersects with stakeholder involvement, where patient and public engagement can improve the trial design and thus the trial outcomes' relevance and applicability.

## Implications

According to the results from the literature review, few studies explicitly defined efficiency in the context of trials and no effort has been made to develop a unified and agreed definition for trial efficiency. Linguistically, 'efficiency' is defined as "the production of the desired effects or results with minimum waste of time, effort, or skill" [79]. This definition shares similarities with those from the literature (S2 Table), wherein the outstanding characteristic corresponds to the balance between the inputs (e.g. resources) and the outputs (e.g. the objectives of the trial). Nevertheless, these interpretations are often narrowly tailored. In this study we hoped to offer a holistic view that captures the nuances and complex aspects of trial efficiency and which may benefit policymakers, funders, and researchers in making informed decisions, leading to improved trial implementation and patient care. Enhancing efficiency was emphasised in the UK Department of Health and Social Care's 2022–2025 strategic plan for clinical research [80]. As of the drafting of this paper, the U.S. Food and Drug Administration is announcing the updated recommendations for good clinical practices advocating for greater efficiency in trials by modernising both design and conduct [81]. Therefore, it is evident that our study is timely, positioning the urgency of comprehensively understanding trial efficiency.

## Strengths and limitations

Drawing on both literature review and expert opinion, our study followed a rigorous approach to develop a conceptual framework of trial efficiency. We included a wide range of experts in trial communities including members of the public, enhancing the comprehensiveness and richness of our study. Nevertheless, nine participants did not respond to the scoring round,

which could have introduced potential biases in reaching a consensus or perhaps missed subtle distinctions regarding the significance of certain trial elements. However, given the diverse range of participants who did engage, coupled with the triangulation with existing literature, this non-response is not expected to significantly impact the overall validity and comprehensiveness of our Delphi findings.

While we have sought to delineate each construct and building block distinctly, we acknowledge the potential for different interpretations of qualitative data. The interplay between the identified themes is likely to be more intricate, reflecting the complex nature of trial efficiency. Future research could delve deeper into this interplay to unravel the connections.

The 'trial efficiency pentagon', emerging as a novel concept from this study, is a promising tool for assessing trial efficiency (proactively and retrospectively). For example, it could be developed to support group discussions and/or calibrated as an evaluation instrument to measure the efficiency of a trial. However, it is limited by lacking robust theoretical foundation. To elucidate, while we've pieced together insights and perspectives to shape the pentagon, we have not rooted it in any established theory or conceptual model. This could mean that certain fundamental aspects of trial efficiency might be overlooked or not holistically represented. In the future, we aspire to hone the pentagon into an evidence-based, theory-informed tool and we welcome insights from our readers and remain open to potential collaborations to its further development.

## Conclusions

This is the first attempt to dissect the concept of trial efficiency into theoretical constructs. In this pursuit of understanding, we are not only unravelling the complexities of trial efficiency but also laying the groundwork for evaluating the efficiency of an individual trial or a trial system in the future.

## Supporting information

**S1 Fig. PRISMA flowchart.**
(DOCX)

**S2 Fig. Trial process in general.**
(DOCX)

**S1 Table. Literature review inclusion and exclusion criteria.**
(DOCX)

**S2 Table. Efficiency definitions/explanations in the literature.**
(DOCX)

**S3 Table. Scoring round stratified results.**
(DOCX)

**S4 Table. Scoring round exemplar quotes related to potential overlaps among the four constructs.**
(DOCX)

**S1 File. Open round questionnaire.**
(DOCX)

## Acknowledgments

We thank Prof. Shaun Treweek for his insightful discussion on trial efficiency, which has largely inspired this work. We thank Ann Thomson, Senior Trial Manager at Queen Mary University of London's Pragmatic Clinical Trials Unit, for her valuable discussions and insights into the trial process. Our thanks also go to the Health Research Board - Trials Methodology Research Network for their assistance in promoting our Delphi study through their email newsletter. We acknowledge the support of the UKCRC Registered CTU Network. The views expressed are those of the author(s) and not of the UKCRC or its members. We are immensely thankful to all participants of the Delphi study rounds for their invaluable contributions and willingness to share their expertise. We have received consent to acknowledge the following participants by name (with no particular order): Monica Taljaard, Lelia Duley, Sarah Markham, Deb Smith, Catey Bunce, Stephen Brealey, Steff Lewis, Laura Miller, Jacqueline French, Fiona Hogarth, Gail Holland, Nikki Totton, Nick Kisengese, Joanne Haviland, Matthew Burns, Richard Hooper, Claire Ayling, Catherine Arundel, Ines Rombach, Seonaidh Cotton, Paula Kareclas. Lastly, we appreciate the reviewer's comments, which have been instrumental in enhancing the development of the conceptual framework.

## Author Contributions

**Conceptualization:** Charis Xuan Xie, Anna De Simoni, Sandra Eldridge, Hilary Pinnock, Clare Relton.

**Data curation:** Charis Xuan Xie.

**Formal analysis:** Charis Xuan Xie.

**Funding acquisition:** Charis Xuan Xie.

**Investigation:** Charis Xuan Xie.

**Methodology:** Charis Xuan Xie, Anna De Simoni, Sandra Eldridge, Hilary Pinnock, Clare Relton.

**Project administration:** Charis Xuan Xie.

**Resources:** Charis Xuan Xie.

**Software:** Charis Xuan Xie.

**Supervision:** Anna De Simoni, Sandra Eldridge, Hilary Pinnock, Clare Relton.

**Validation:** Anna De Simoni, Sandra Eldridge, Hilary Pinnock, Clare Relton.

**Visualization:** Charis Xuan Xie.

**Writing – original draft:** Charis Xuan Xie.

**Writing – review & editing:** Charis Xuan Xie.

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
