## [Decision Letter · Decision Letter 0]

14 Feb 2024

PONE-D-23-40114Development of a conceptual framework for defining trial efficiencyPLOS ONE

Dear Dr. Xie,

Thank you for submitting your manuscript to PLOS ONE. After careful consideration, we feel that it has merit but does not fully meet PLOS ONE’s publication criteria as it currently stands. Therefore, we invite you to submit a revised version of the manuscript that addresses the points raised during the review process.

We look forward to receiving your revised manuscript.

Kind regards,

Germain Honvo, Ph.D.

Academic Editor

PLOS ONE

Journal Requirements:

Reviewers' comments:

Reviewer's Responses to Questions

**Comments to the Author**

1. Is the manuscript technically sound, and do the data support the conclusions?

Reviewer #1: Yes

Reviewer #2: Yes

2. Has the statistical analysis been performed appropriately and rigorously? 

Reviewer #1: Yes

Reviewer #2: N/A

3. Have the authors made all data underlying the findings in their manuscript fully available?

Reviewer #1: Yes

Reviewer #2: No

4. Is the manuscript presented in an intelligible fashion and written in standard English?

Reviewer #1: Yes

Reviewer #2: Yes

5. Review Comments to the Author

Reviewer #1: This is a fascinating and potentially very valuable study. By deploying a Delphi e-study, the authors achieved consensus on four theoretical constructs for defining trial efficiency, and on five empirical building blocks essential for

considering trial efficiency. This research will contribute to better evaluation of trial efficiency.

Reviewer #2: Dear Authors,

Well done on the piece of work! The authors have provided a nice framework to define efficiency in the context of trial design and conduct. However, there are a few comments/ suggestions from my end to make the paper accessible to a greater audience.

1. The most interesting question after reading the paper has been: as the authors identified four constructs and 5 building blocks of the trial efficiency concept, what is the interplay between them? As in, how each block will fall under the defined constructs? A researcher might be interested if he/she is looking for a particular kind of efficiency, which building blocks does he/she needs to consider? I would also be interested in looking at the potential overlap between these constructs (if there are any, which I think there might be). Any diagrammatic summary would be nice.

2. Please include the e-delphi questionnaire that have been used. It confuses me if Figure 1 summarise all the questions asked in the survey. As mentioned there, “Question 1. How do you define efficiency within the context of trials to improve healthcare?” How was this recorded? It will give the readers a clearer picture if the questionnaire is attached.

3. I think there might be some overlap between the identified themes from a literature review, for example: Efficient trial designs (as mentioned in the description e.g. Adaptive designs, Master protocols etc) are mostly more sophisticated statistical trial designs instead of a simple RCT (where the paradigm looks like design -> conduct -> analysis). In that case why it is not different from statistical efficiency or operational/economical efficiency (which the designs typically target to optimise)? Why can’t it not fall under them?

4. Please include some description of the figures in the text/caption of the figure. I am still unsure of what figure 2 describes. It is not clear whether it reports some sort of frequency or ranking or percentage.

5. A quick clarification: what do the authors mean by member of public? Are they trial participants?

A small typo in the appendix, all the inter-percentile ranges in IPRAS are mentioned as intercentile ranges.

Hope this helps! Good luck!

6. PLOS authors have the option to publish the peer review history of their article (what does this mean?). If published, this will include your full peer review and any attached files.

Reviewer #1: **Yes: **Dr Sarah Markham

Reviewer #2: No

---

## [Author Response · Author response to Decision Letter 0]

22 Mar 2024

Review Comments to the Author

Reviewer #1: This is a fascinating and potentially very valuable study. By deploying a Delphi e-study, the authors achieved consensus on four theoretical constructs for defining trial efficiency, and on five empirical building blocks essential for considering trial efficiency. This research will contribute to better evaluation of trial efficiency.

We appreciate your recognition of our study's value and its potential impact on the field. 

Reviewer #2: Dear Authors,

Well done on the piece of work! The authors have provided a nice framework to define efficiency in the context of trial design and conduct. However, there are a few comments/ suggestions from my end to make the paper accessible to a greater audience.

1. The most interesting question after reading the paper has been: as the authors identified four constructs and 5 building blocks of the trial efficiency concept, what is the interplay between them? As in, how each block will fall under the defined constructs? A researcher might be interested if he/she is looking for a particular kind of efficiency, which building blocks does he/she needs to consider? I would also be interested in looking at the potential overlap between these constructs (if there are any, which I think there might be). Any diagrammatic summary would be nice

Thank you for highlighting this crucial aspect of our study. 

The potential overlaps between the constructs have been discussed in the original manuscript in the discussion section, on pages 21 and 22. 

Regarding the relationships between the constructs and the building blocks, we initially distinguish between the constructs and the building blocks to illustrate a layered understanding of trial efficiency. The four constructs were conceptualised to provide a broad, overarching view of efficiency within healthcare trials. In contrast, the five building blocks were identified as the essential, practical components that operationalise efficiency in real-world settings.

However, recognising the relationships among these identified themes is indeed essential in the theory-building process. As such, we have added the following descriptions in the revised discussion, on page 23. 

“There is a layered connection between the constructs and the building blocks: the four constructs were conceptualised to provide a broad, overarching view of efficiency within healthcare trials. In contrast, the five building blocks were identified as the essential, practical components that operationalise efficiency in real-world settings. In addition to this relationship, we suggest that for a comprehensive understanding, each efficiency construct takes into account all five building blocks. For instance, while it may seem apparent that scientific efficiency is closely linked with trial design, focusing on how the study is conceptualised to ensure methodological soundness; it also intersects with stakeholder involvement, where patient and public engagement can improve the trial design and thus the trial outcomes' relevance and applicability. Figure 5 illustrates the potential interplay between the theoretical constructs and the empirical building blocks.”

And in light of your suggestion, we have included an updated conceptual framework of trial efficiency (figure 4 on page 21) to visually outline the theoretical constructs, building blocks and their relationship. This diagrammatic representation is part of our ongoing effort to refine our conceptual framework and enhance its applicability and understanding:

Figure 4. The conceptual framework of trial efficiency.

Footnote: The outer blue circle outlines four theoretical constructs of trial efficiency, which are Scientific Efficiency, Statistical Efficiency, Operational Efficiency and Economic Efficiency. At its core, the inner pentagon outlines five empirical building blocks: Superstructure, Stakeholders, Infrastructure, Trial Process, and Trial Design. The cyclical arrows indicate the necessity for a balanced consideration of each building block within each construct to optimise trial efficiency.

2. Please include the e-delphi questionnaire that have been used. It confuses me if Figure 1 summarise all the questions asked in the survey. As mentioned there, “Question 1. How do you define efficiency within the context of trials to improve healthcare?” How was this recorded? It will give the readers a clearer picture if the questionnaire is attached.

The e-Delphi questionnaire has been added to the supplementary file 1. Question one was recorded in free text. The following changes have been made to the manuscript on page 8, method section:

“In the open round, we invited panellists to share their thoughts on 1) their understanding of trial efficiency and 2) the most efficient or inefficient aspects they have encountered in the trials they have conducted or in which they have participated. These questions were designed as free text to encourage detailed, narrative responses. (see S1 File 1 for the questionnaire).”

3. I think there might be some overlap between the identified themes from a literature review, for example: Efficient trial designs (as mentioned in the description e.g. Adaptive designs, Master protocols etc) are mostly more sophisticated statistical trial designs instead of a simple RCT (where the paradigm looks like design -> conduct -> analysis). In that case why it is not different from statistical efficiency or operational/economical efficiency (which the designs typically target to optimise)? Why can’t it not fall under them?

Thank you for raising the point about the overlaps among the four constructs. We provide the following rationale for categorising trial design into scientific efficiency:

The theme synthesis was partially based on the previous discussions about scientific efficiency and operational efficiency [1], in which the authors argued that innovative clinical trial designs (such as adaptive studies) are integral to scientific efficiency. 

[1] Kelly D, Spreafico A, Siu LL. Increasing operational and scientific efficiency in clinical trials. Br J Cancer. 2020;123(8):1207-8.

While these designs inherently enhance statistical efficiency through sophisticated design features, they are considered distinct due to their broader impact on the scientific process. They extend beyond statistical considerations to influence the trial's methodological soundness, ethical conduct, and the potential for quicker patient benefit.

Additionally, operational efficiency, as addressed in our review, involves the effective management and organisation of trial processes and procedures. Sophisticated trial designs aim to improve these aspects but do not encapsulate the entirety of operational considerations.

Economic efficiency, distinct yet linked, focuses on the cost implications of conducting a trial. Innovative trial designs can influence costs but are one of many factors that contribute to the overall economic efficiency of a trial.

By delineating these designs as a separate entity, we aim to emphasise their unique contribution to trial efficiency. They are a part of the broader constructs but deserve individual recognition for their specific impacts. As such, we clarified this interplay in the discussion section of the original manuscript, on page 22: 

“Overall there was no disagreement over the four constructs ……thus offering a nuanced theoretical framework for its comprehension.”

Nonetheless, we recognise that interpretations of qualitative data can vary, and to account for this, we have included the following statement in the limitations section:

“While we have sought to delineate each construct and building block distinctly, we acknowledge the potential for different interpretations of qualitative data. The interplay between the identified themes is likely to be more intricate, reflecting the complex nature of trial efficiency. Future research could delve deeper into this interplay to unravel the connections.”

4. Please include some description of the figures in the text/caption of the figure. I am still unsure of what figure 2 describes. It is not clear whether it reports some sort of frequency or ranking or percentage.

Thank you, we have added the following note to figure 2:

“Figure footnote: The x-axis represents the frequency of responses.”

5. A quick clarification: what do the authors mean by member of public? Are they trial participants?

Thank you for seeking further clarification. In our study, 'trial participants' are referred to as individuals who have directly participated in clinical trials (e.g. receiving interventions or being part of control groups), offering invaluable insights from their lived experiences. This is to separate the role of 'members of the public', who may not have personally participated in clinical trials, but may have helped in patient engagement and/or assisted in developing study designs. 

In light of this comment, we have clarified it in the revised manuscript under the section of “Patient and public involvement” on page 9:

“In this study, members of the public (n=4) (including two who had participated in trials) were invited to share their thoughts, participate in the ranking process, provided with the outcomes of each round upon completion.”

6. A small typo in the appendix, all the inter-percentile ranges in IPRAS are mentioned as intercentile ranges.

Thank you for spotting this typo. They have been corrected accordingly.

---

## [Decision Letter · Decision Letter 1]

8 May 2024

Development of a conceptual framework for defining trial efficiency

PONE-D-23-40114R1

Dear Dr. Xie,

We’re pleased to inform you that your manuscript has been judged scientifically suitable for publication and will be formally accepted for publication once it meets all outstanding technical requirements.

Kind regards,

Germain Honvo, Ph.D.

Academic Editor

PLOS ONE

Additional Editor Comments (optional):

Reviewers' comments:

Reviewer's Responses to Questions

**Comments to the Author**

1. If the authors have adequately addressed your comments raised in a previous round of review and you feel that this manuscript is now acceptable for publication, you may indicate that here to bypass the “Comments to the Author” section, enter your conflict of interest statement in the “Confidential to Editor” section, and submit your "Accept" recommendation.

Reviewer #1: All comments have been addressed

Reviewer #2: (No Response)

2. Is the manuscript technically sound, and do the data support the conclusions?

Reviewer #1: Yes

Reviewer #2: (No Response)

3. Has the statistical analysis been performed appropriately and rigorously? 

Reviewer #1: Yes

Reviewer #2: (No Response)

4. Have the authors made all data underlying the findings in their manuscript fully available?

Reviewer #1: Yes

Reviewer #2: (No Response)

5. Is the manuscript presented in an intelligible fashion and written in standard English?

Reviewer #1: Yes

Reviewer #2: (No Response)

6. Review Comments to the Author

Reviewer #1: I am happy with the revisions made by the authors. This conceptual paper will make a positive contribution to the literature.

Reviewer #2: (No Response)

7. PLOS authors have the option to publish the peer review history of their article (what does this mean?). If published, this will include your full peer review and any attached files.

Reviewer #1: No

Reviewer #2: No

---

## [Editor Report · Acceptance letter]

14 May 2024

PONE-D-23-40114R1 

PLOS ONE

Dear Dr. Xie, 

I'm pleased to inform you that your manuscript has been deemed suitable for publication in PLOS ONE. Congratulations! Your manuscript is now being handed over to our production team.

Kind regards, 

on behalf of

Dr. Germain Honvo 

Academic Editor

PLOS ONE